# Identification and Tracking of Vehicles between Multiple Cameras on Bridges Using a YOLOv4 and OSNet-Based Method

**DOI:** 10.3390/s23125510

**Published:** 2023-06-12

**Authors:** Tao Jin, Xiaowei Ye, Zhexun Li, Zhaoyu Huo

**Affiliations:** Department of Civil Engineering, Zhejiang University, Hangzhou 310058, China; cetaojin@zju.edu.cn (T.J.);

**Keywords:** structural health monitoring, deep learning, temporal–spatial distribution, vehicle loads, vehicle identification

## Abstract

The estimation of vehicle loads is a rising research hotspot in bridge structure health monitoring (SHM). Traditional methods, such as the bridge weight-in-motion system (BWIM), are widely used but they fail to record the locations of vehicles on the bridges. Computer vision-based approaches are promising ways for vehicle tracking on bridges. Nevertheless, keeping track of vehicles from the video frames of multiple cameras without an overlapped visual field poses a challenge for the tracking of vehicles across the whole bridge. In this study, a method that was You Only Look Once v4 (YOLOv4)- and Omni-Scale Net (OSNet)-based was proposed to realize vehicle detecting and tracking across multiple cameras. A modified IoU-based tracking method was proposed to track a vehicle in adjacent video frames from the same camera, which takes both the appearance of vehicles and overlapping rates between the vehicle bounding boxes into consideration. The Hungary algorithm was adopted to match vehicle photos in various videos. Moreover, a dataset with 25,080 images of 1727 vehicles for vehicle identification was established to train and evaluate four models. Field validation experiments based on videos from three surveillance cameras were conducted to validate the proposed method. Experimental results show that the proposed method has an accuracy of 97.7% in terms of vehicle tracking in the visual field of a single camera and over 92.5% in tracking across multiple cameras, which can contribute to the acquisition of the temporal–spatial distribution of vehicle loads on the whole bridge.

## 1. Introduction

Bridge structures are critical components of transportation infrastructures that contribute to the smoothness of traffic flow. However, as time goes by, the safety of in-service bridges is challenged by the effects of multiple factors, especially vehicle loads [1,2,3,4]. On one hand, the load standards for the design of bridges were determined decades ago yet there are more vehicles and heavier vehicles on the bridges nowadays. On the other hand, eccentric loading of bridges induced by the unilateral passage of heavy vehicles, such as trucks and flat cars, will also lead to damage to bridge components [5,6]. Therefore, the acquisition of vehicle loads on the bridges is vital to decisions regarding maintenance for structural safety.

Thanks to the development of sensing techniques and data processing algorithms, SHM-based methods have been proposed to detect vehicle loads on bridges. Among the many kinds of techniques for sensing vehicle loads, the bridge weight-in-motion system (BWIMs) proposed by Moses [7] is preferred for practical application by those in the industrial community [8,9]. The BWIMs applies sensors to capture bridge stress and strain, and analyzes its dynamic strain responses to restore vehicle information, including vehicle load, speed, number of axles, etc. [10]. Its high precision and broad applicability has attracted many scholars. Wu et al. [11] developed an encoder–decoder structure called BwimNet to identify the properties of moving vehicles. The model is multi-target and can detect axle number, speed, weight, and wheelbases simultaneously. After decades of development, the BWIMs present excellent performance and has been installed on many major bridges. However, the BWIMs are fixed in a certain section and can detect the vehicle loads only when vehicles are passing over the sensors embedded under the bridge deck [12]. Thus, the temporal–spatial distribution of vehicle loads on bridges is not available when using the BWIMs.

In recent years, many researchers have adopted vision-based methods to locate vehicles due to the rapid development of computer vision techniques [13,14,15,16,17,18]. Chen et al. [19] proposed a Gaussian Mixture Model (GMM) and a shadow removal method-based approach to detect and track vehicles through CCTV devices. Chen et al. [20] established a real-time vehicle detection and counting method based on single shot detection (SSD) and reached an accuracy of 99.3%. The vehicles were classed into six groups, including cars, taxis, vans, trucks, motorbikes, and buses. Harikrishnan et al. [21] put forward a bounding box algorithm to locate vehicles, which was estimated with two-dimensional binary histogram projection profile (2D-BHPP) algorithm. Zhang et al. [22] developed a vehicle detection algorithm based on the Faster region-based convolutional neural network (Faster R-CNN), and the Zeiler and Fergus model (ZF). The method was applied to automatically detect vehicle types, number of axles, and length for the temporal–spatial information of vehicles on bridges. The computer vision-based vehicle detection approaches could obtain the temporal–spatial distribution of vehicle loads in the field of a single camera which covers a limited portion of the whole bridge. When the visual fields of multiple cameras are not continuous, the shapes of the same vehicle in video frames of different cameras will be quite different, which challenges the detection of the vehicles along the whole bridge, as illustrated in Figure 1.

To overcome the existing problem, Chen et al. [23] applied feature and area-based approaches to re-identify vehicles between multiple cameras. Edge detection was applied to extract the features of vehicles, and a template matching algorithm was used to track vehicles. The temporal–spatial distribution model of traffic loads on Hangzhou Bay Bridge was obtained. Yet, the template matching algorithm is sensitive to the quality of the captured video images. Dan et al. [24] used Kalman Filter to track a vehicle and calculated the time it should appear in the visual field of the next camera. The best matched one would be considered as the same vehicle, but it could make false predictions when the distances between the adjacent vehicles are small.

The vehicle re-identification between multiple cameras is a hot field in computer vision [25,26,27]. The number of re-identification studies has grown in number, aiming to solving the challenge of matching objects across different cameras when the primary hallmark, such as the face or plate number, is unrecognized. Many neural network models that focus on the re-identification issue have been put forward [28,29,30], and due to the distinctiveness of vehicles, additional information has been applied for precise re-identification. The Siamese-CNN + Path-LSTM model proposed by Shen et al. [31] takes the vehicle path into account. These studies mainly concentrate on modifying models and methods to improve performance in the existing dataset, such as VeRi-776 [32], CompCars [33], and VERI-Wild [34].

Inspired by the re-identification method, a YOLOv4 and OSNet-based method for identification of the temporal–spatial distribution of vehicle loads on bridges was proposed in this study. It includes a YOLOv4-based vehicle detection module and an OSNet-based feature extraction and re-identification module. A dataset with 25,080 images related to 1727 vehicles was established to train the OSNet and a field validation experiment was conducted to test the proposed method for evaluation of robustness and reliability.

## 2. Framework of the Proposed Method

The proposed vehicle detection, tracking, and re-identification method based on the YOLOv4 neural network and the OSNet is shown in Figure 2. The proposed method is mainly composed of two modules, one is for vehicle detection from the video images and the other is for re-identification of the same vehicle from the video images captured by different cameras.

In the first module, YOLOv4 is applied to detect and locate vehicles, while the Kalman Filter and a modified IoU-based tracking method are used to track the same vehicle between adjacent frames captured by the same camera. Every vehicle with corresponding information regarding location and time will be stored and post-processing is adopted to suppress the interference of incorrect detection. In addition, a novel algorithm for position correction was proposed to clear the lanes of vehicles. The second module is a vehicle re-identification module, which adopts OSNet, to extract features of vehicle images. After that, a re-ranking method was introduced to amend the Euclidean distance between image features, and the Hungary algorithm proposed by Munkres [35] was adopted to match the same vehicle from images captured by multiple cameras without an overlapped visual field.

## 3. Vehicle Detection and Tracking with a Single Camera

Vehicle detection and tracking based on a single camera is the foundation of the proposed method. The flowchart for processing the video frames from the same camera for vehicle detection and tracking is shown in Figure 3. Vehicle detection is conducted by the YOLOv4 and the tracking of vehicles is realized with a modified IoU-based method.

### 3.1. Architecture of YOLOv4

The YOLOv4 proposed by Bochkovskiy et al. [36] has been adopted by many researchers for satisfactory performance [37,38] and has been utilized in this study. Shown in Figure 4, the architecture of YOLOv4 contains three parts, CSPDarknet53 as the backbone, SPP + PAN as the neck, and YOLOv3 as the head. In the SPP, there are three pooling channels with different kernel sizes, which are 5 × 5, 9 × 9 and 13 × 13.

The CSPDarknet53 adopted a cross-stage feature fusion strategy named Cross Stage Partial (CSP) proposed by Wang et al. [39]. The feature map is divided into two parts, one part goes through convolution layers and the other passes through a shortcut and concatenates with the former. Thus, it reduces computations without degradation of detection ability. CSPDarknet53 consists of Darknet53, the backbone of YoLOv3, and CSP to achieve fast and precise detection. The Spatial Pyramid Pooling (SPP) method proposed by He et al. [40] and the Pixel Aggregation Network (PAN) proposed by Wang et al. [39] are combined to fuse features at different scales. They allow YOLOv4 to transmit features from various layers and benefits the feature extraction. The prediction module of YOLOv4 applies three scales to detect objects with different sizes, and outputs their positions, categories, and confidences.

Moreover, a batch of methods are tested in YOLOv4 to achieve a higher level of detection. These methods cover activations, bounding box regression loss, data augmentation, and so on. Among them, Distance-IoU (DIoU) contributes a lot to reducing the possibility of low recall, and has a higher potential in vehicle detection. The previous non-maximum suppression (NMS) algorithm only adopts intersection over union (IoU) to remove redundant bounding boxes and retain the one that is most possible. In response to this, DIoU takes the distance of box centers into consideration along with IoU, and reduces the mistaken elimination, as shown in Figure 5.

The original resolution of the surveillance videos is 1080 × 1920, which does not suit the requirement of YOLOv4. In order to achieve a balance between speed and accuracy, the frames were resized into 512 × 896 resolutions, which maintains the original aspect ratio as much as possible.

### 3.2. Modified IoU-Based Tracking Method

Tracking vehicles between adjacent frames meets the challenges of multiple vehicles, missing detection, and false-positive detection. In order to overcome these challenges, Chen et al. [20] applied a bounding box distance between the box center of consecutive sequence of frames to implement vehicle tracking. However, the neglect of vehicle movement limits its recognition capability, and for a bounding box, there is only one point used to track, which will be often disturbed by bounding boxes of different sizes. Zhou et al. [16] adopted the Kalman filter to predict vehicle positions in the current frame depending on the former tracks, and applied the predicted box to track vehicles. Previous research mainly focused on spatial information of vehicles while appearance features of the vehicles in the bounding boxes can make a contribution to vehicle tracking as well [41]. Therefore, a modified IoU-based tracking method which takes appearance into consideration was proposed, as shown in Figure 6.

During the processing of vehicle detection via YOLOv4, a batch of rectangular object proposals with corresponding scores and categories were obtained, and these bounding boxes will be compared with the bounding boxes predicted in the last frame, as Figure 7a shows. IoU is used to estimate their overlap ratio and calculated as Figure 7b. Each bounding box of the current frame will obtain a list of IoU values which represents its overlapping ratio with the last determined bounding boxes, and the maximum will be chosen as the most likely one for multiple vehicles. Matched results are classified into three classes, the strong match, the weak match, and the excluded match. A strong match whose IoU value is over 0.4 means two bounding boxes are likely to represent the same vehicle. Afterwards, the corrected detection result based on prediction will be calculated and appended to the track of the detected vehicle. Moreover, an additional bounding box will be predicted for tracking in the next frame. An excluded match means the detecting bounding box does not overlap with any of the predicted boxes and suggests a new vehicle might appear in this frame. Then, a new series number will be generated to refer to this vehicle, and the exterior information extracted by the OSNet will be bound to this vehicle. It should be explained that every bounding box in the first frame is regarded as an excluded match.

The main difficulty comes from the weak match whose IoU value is between 0 and 0.4. External information is adopted for accurate judgment. The current vehicle image will be input into the OSNet and the characteristics of the appearance of vehicles will be obtained. Then, Euclidean distance between the characteristics of the appearance of two vehicle images could be calculated. Due to the similar direction of the visual field in a single camera, the images of the same vehicle usually share a similar appearance, and their Euclidean distance would be significantly smaller than that of different vehicles. With this approach, the classification of weak matches is obtained. The tracking method is a recurrent procedure which will keep working until all the video frames are processed as shown in Figure 3.

### 3.3. Kalman Filter

The Kalman filter is a recursive method that estimates the state of the target object combined with prediction and detection results [42]. An 8-dimensional state space z for prediction is established as follows.
(1)$$x=\left[m,n,a,h,m^{\prime },n^{\prime },a^{\prime },h^{\prime }\right]$$
where *m* and *n* refer to the bounding box center position, *a* stands for the aspect ratio, and *h* is the height of the bounding box. The next couple of terms represent their derivatives, respectively. Forecasts to draw a predicted box are performed by using the following equation:(2)$${x_{i}}^{\prime }=Ax_{i-1}$$
(3)$${P_{i}}^{\prime }=AP_{i-1}A^{T}+Q$$

The matrix *A* relates the state at the previous time step to the current step, and the state covariance matrix of the last time step *P_i−_*_1_ is used to calculate that in the next step *P_i_*. *Q* stands for indeterminacy of state.

Updating operation works after prediction:(4)$$K_{i}={P_{i}}^{\prime }H^{T}{\left(H{P_{i}}^{\prime }H^{T}+R\right)}^{-1}$$
(5)$$x_{i}={x_{i}}^{\prime }+K\left(z_{i}-H{x_{i}}^{\prime }\right)$$
(6)$$P_{i}=\left(I-K_{i}H\right)P_{i}H^{\prime }$$

The first step of updating is to calculate the Kalman gain *K*, and the measurement matrix *H* relates measurement to state. *R* stands for noises of devices. The second step is to gain the revised state estimate based on the measurement *z* and predicted state *x_i_*^′^. Finally, a posteriori error covariance estimate is obtained.

### 3.4. Adaptive Lane Division Method

There are a lot of studies on conversion from the camera coordinate system to the world coordinate system [17,43]. However, the center or the bottom of the bounding box cannot represent the position of the target vehicle precisely (Figure 8), and may lead to the error distinction.

In this section, an alternative approach for lane division was proposed. A batch of bounding boxes on pilot run of YOLOv4 with no tracking operation can be obtained, and their midpoints of rectangular bottom lines can be noted, as shown in Figure 9. The dots converge into six lines, each representing their corresponding lane. Red lines shown in Figure 8 are the ground-truth axes of lanes and they do not match dots completely. The yellow lines are the centers of the dots shown in Figure 9. Therefore, we applied a two-step method, firstly (i) classify dots depending on the distance from the points to the base-lines and then (ii) every group is used to fit a new base-line for the least variance.

The method will be repeated until the result is almost unchanged. Compared with the initial lines in red, yellow lines are more suitable for lane division, especially for the one far away from the camera.

## 4. OSNet-Based Vehicle Re-Identification

The OSNet is a convolutional neural network (CNN) focusing on re-identification issues. The OSNet can capture different spatial scales, and integrate these scales as output.

### 4.1. The Bottleneck of the OSNet

Figure 10 shows the bottleneck of the OSNet. It is evident that the bottleneck contains four paths with different numbers of convolution layers, thus various scale features are obtained. The numbers are the amount of lite convolutions in different paths. Aggregation gate (AG) refers to the unified aggregation gate, which controls weights assigned to different scales. The aggregation gate is novel to others in terms of its ability to learn, which requires less human intervention. The lite convolution separates a standard convolution layer into a pointwise layer and a depthwise layer. In the case that the result of calculation is slightly changed, this operation significantly reduces the amount of calculation. The OSNet network will automatically select different outputs according to the model state. During the training stage, features (when triple loss is adopted) and categories will be output, and during the test stage, only features will be output.

### 4.2. Establishment of Dataset

A vehicle re-identification dataset is established to train the OSNet. Pictures in the dataset were captured as explained in Section 2 with the exception of the classification of weak matches. In the process of dataset establishment, all the weak matches were considered as excluded matches, and two members of this study checked them manually. After making sure the vehicle images of a single camera were properly classified, vehicle images from different camera frames were labeled. Thus, every vehicle in the established dataset was captured by at least two cameras and all the pictures of it share the same label. Finally, 25,080 images of 1727 vehicles were collected and labeled, for each of which there was a camera marking. Figure 11 shows examples of the established dataset.

The dataset is divided into two subsets for training and testing, respectively. The training set contains 1727 vehicles with 25,080 images and the testing set has 200 vehicles with 2891 images. Additionally, open-sourced datasets VeRi-776 and VERI-Wild were utilized for a comparison study.

### 4.3. Data Processing

For this study, the size of the input images was set as 256 × 256 resolutions, which fits the shape of vehicle images better. In order to improve the OSNet’s robustness, the images were resized into 288 × 288 resolutions and randomly cropped to 256 × 256 resolutions. Horizontal flip at a possibility of 0.5 was adopted also for augment processing in this study.

### 4.4. Evaluating Indices

In this study, the Rank-1 and the mean average precision (mAP) index were adopted as the evaluation of vehicle identification performance. Three other re-identification models including Res50-dim [44], PCB [45] and HA-CNN [46] were applied as contrasts. In the training process, a query picture was input, and the neural network searched the picture belonging to the same category from the gallery images and ranked them according to their probability. Rank-1 represents the rate that the picture with the highest probability and query one are indeed the same vehicle. The mAP was used to denote the precision of the neural network as follows.
(7)$$mAP=\frac{\sum_{k=1}^{C}AP_{k}}{C}$$
(8)$$AP=\frac{\sum_{i=1}^{n}{\mathrm{Precision}}_{i}}{n}$$
(9)$$Precision_{i}=\frac{i}{Position_{i}}$$

Every registered vehicle has a couple of pictures with the same label, and in Equation (8), the *n* stands for the number of this category, and *Precision_i_* is the precision of the *i*th picture of the registered vehicle.

### 4.5. Hyper-Parameter

Adam proposed by Kingma and Ba [47] is adopted as the optimizer in this paper. The cross entropy loss and triplet loss were adopted for weight updating. The cross entropy loss is calculated as follows
(10)H(p,q)=−∑(p(x)logq(x))
where *H*(*p*, *q*) is the cross entropy loss applied to updating the neural network weights along with triplet loss. *p*(*x*) represents the label of input vehicle image, and *q*(*x*) is the output possibility of if the input vehicle is the same one as the label.

Triplet loss was proposed for face re-identification by Schroff et al. [48]. It aims to shorten the distance of features from the same category and enlarge the distance from various types. It is defined by
(11)L=max(d(a,p)−d(a,n)+margin,0)
where *a* means anchor, *p* stands for positive sample, and *n* stands for negative sample. *margin* is a constant usually defaulted as 0.3.

The hyperparameter is summarized in Table 1 and was utilized to train four different models.

### 4.6. Training Results

The OSNet was trained in a workstation, and its hardware and software are listed in Table 2. Torchreid is a software library based on Pytorch, and it allows convenient training and evaluation of re-identification models.

Figure 12 shows the evaluation indicators of diverse models, and obviously, the OSNet surpasses other methods by a clear margin.

## 5. Methods for Improvement of Vehicle Re-Identification

### 5.1. Reranking Method

Vehicle features can be extracted by the OSNet, and the Euclidean distance can be calculated by Equation (12)
(12)$$d\left(p,q\right)=∥f_{p}-f_{q}{∥}_{2}^{2}$$
where *p* and *q* stand for two vehicle images, and *f_p_*, *f_q_* represent their features, respectively. We apply the inverse as an indicator.

A reranking method presented by Zhong et al. [49] was adopted in this step. Reranking is a post-processing that helps to improve the initial ranking result without requiring extra labels or training data [50].
(13)$$N\left(p,k\right)=\left\{g_{1},g_{2},g_{3},...,g_{k}\right\},\left|N\left(p,k\right)\right|=k$$

As Equation (13) shows, we draw a set of pictures {*g*_1_, *g*_2_, *g*_3_, …, *g*_k_} as the k-nearest neighbors *N*(*p*, *k*). It presents the initial order of Euclidean distances between gallery pictures and the query picture *p*. Then, we apply Equation (14) to obtain the k-reciprocal nearest neighbors *R*(*p*, *k*), a subset of *N*(*p*, *k*), whose members are more related to the query picture *p* and consider *p* as their k-nearest neighbors.
(14)$$R\left(p,k\right)=\left\{g_{i}|g_{i}\in N\left(p,k\right)\&p\in N\left(g_{i},k\right)\right\},\left|N\left(p,k\right)\right|=k$$

The set *R*(*p*, *k*) is the bidirectional k-nearest neighbors and effectively excludes false-positive pictures. In addition, a method was adopted to improve recall rate by Equation (15):(15)$$\begin{array}{cc} & R*\left(p,k\right)=R\left(p,k\right)⋃R\left(1,\frac{1}{2}k\right)\\ & s.t.|R\left(p,k\right)⋂R\left(1,\frac{1}{2}k\right)|\geq \frac{2}{3}\left|R\left(1,\frac{1}{2}k\right)\right|\\ & \forall q\in R(p,k) \end{array}$$
where *R**(*p*, *k*) consists more positive samples, which are k/2-nearest neighbors of candidates in *R*(*p*, *k*) and are likely to represent the same vehicle as the query picture. The |$R(1,\frac{1}{2}k)$| denotes the number of members in the set $R(1,\frac{1}{2}k)$.

A pairwise distance *d*_J_(*p*, *g_i_*) called Jaccard Distance between query picture *p* and gallery g is calculated by Equation (16) when *g_i_* belongs to *R**(*p*, *k*) otherwise is set to 0. It is a new index that stands for the relationships of similarity between pictures.
(16)$$V_{{}_{p,g_{i}}}=e^{-d(p,g_{i})}s.t.g_{i}\in R^{*}\left(p,g_{i}\right)$$
(17)$$\begin{array}{c} d_{J}\left(p,g_{i}\right)=1-\frac{\sum_{j=1}^{N}\min\left(V_{p,g_{i}},VL_{g_{i},g_{j}}\right)}{\sum_{j=1}^{N}\max\left(V_{p,g_{i}},VL_{g_{i},g_{j}}\right)} \end{array}$$
*V_p, gi_* is the initial similarity, a numerical value that converted from the initial distance. The final distance is defined by
(18)$$d^{*}\left(p,g_{i}\right)=\left(1-\lambda \right)d_{J}\left(p,g_{i}\right)+\lambda d\left(p,g_{i}\right)$$
where *λ* is a constant that balances the effect of the initial distance *d*(*p, g_i_*) and the Jaccard Distance *d_J_*(*p*, *g_i_*). Finally, re-ranked distances that express more accurate similarities are obtained.

### 5.2. Methods to Reduce the Number of Candidates

In order to reduce the number of candidates, vehicle direction information was taken into consideration since a vehicle almost never turns around on bridges. Similarities between vehicles in different directions can be set to 0. Along with directions, time information was used in this study. A statistical approach was applied to draw time consumed from one monitoring area to another. The time was assumed to be normally distributed, and mean and variance are calculated so as to work out the threshold by function
(19)$$t=\bar{x}+3\sigma$$
where *t* represents a threshold, and $\bar{x}$, σ denotes mean and variance, respectively. Because of the effects combined with time and directions, the number of candidates can be reduced.

### 5.3. Hungary Algorithm to Solve the Assignment Problem

Since the similarities have been computed, the goal is to determine the optimum assignment that maximizes the possibilities, which equals minimizing its opposite number. An approach based on the Hungary algorithm was utilized to match vehicles.

The Hungary algorithm consists of four steps, firstly (i) obtain the minimum value for each row, and subtract it from all the elements in that row; then (ii) every element minus the minimum value in its column; then (iii) use a minimum number of horizontal and vertical lines to cover zeros in the result, and if the number is equal to the number of rows, the positions of zeros are equal to the assignment result; and, finally, (iv) find the smallest element without a line covering it, then subtract it from all the uncovered elements and add it to the elements which are covered twice. Repeat step 3.

Thanks to the Hungary algorithm, the assignment problem for vehicles is solved, and information regarding the same vehicle from different cameras can be merged and output.

## 6. Field Validation of the Proposed Re-Identification Method

The proposed vehicle identification method was verified with video frames from three surveillance cameras on Jiubao Bridge, Hangzhou, China. Figure 13 shows their positions and directions, and Figure 14 demonstrates their views of the same truck. Apparently, camera 1 and camera 2 have completely varying observation directions, and camera 3 shares a similar sight with camera 1. Three ten-minute monitoring videos were obtained from the above cameras for validation.

### 6.1. Hungary Algorithm to Solve the Assignment Problem

After the detecting and tracking process, vehicle tracks and the temporal–spatial distribution of vehicles in the monitoring area were obtained. However, on account of false-positive detection, some wrong trajectory information was included. Figure 15a indicates that this algorithm failed to obtain a good performance in vehicle recognition, and this was due to the low resolution induced by the great distance between the camera and the vehicles. In addition, in a few cases, two bounding boxes or more were recognized to represent the same vehicle as illustrated in Figure 15b.

For precise tracking, two strategies were applied. The first one was to set up a monitoring area and only the vehicles detected in this area would be recorded. This strategy reduces monitoring range but improves robustness. The second one was to set a time limit to exclude the transitory track whose length is shorter. The limit of the time interval is 0.5 s in this study.

These operations ensure that only the correct vehicle trajectories will be stored. After post-processing, we applied accuracy defined by Equation (20) to denote its performance.
(20)$$accuracy=\frac{\mathrm{TP}}{\mathrm{TP}+\mathrm{FN}+\mathrm{FP}}$$

The TP refers to true positive detection, FP and FN refer to false positive detection and false negative detection, respectively. Although the proposed method is sensitive to the sights of cameras, it reached an accuracy over 97.4%, as shown in Table 3. Although the average accuracy is 97.7% which is satisfied, it is not as high as the accuracy achieved by Chen et al. [20] with SSD. The detection distance might be the main influencing factor that the visual field of this study (roughly 180 m) is much larger than that in their investigation. In addition, the performance of the SSD and the YOLO is also slightly different.

At this point, the tracking operations in a single camera were completed, and considering tracking through multiple non-overlapping cameras, another post-processing operation was used. A batch of pictures were captured based on the previous bounding box and the surveillance video. Taking computing time into account, the capturing operation was taken twice per second. Hence, every trajectory had its corresponding information regarding appearance.

### 6.2. Vehicle Re-Identification among Multiple Cameras

The first camera was set as the basic camera, the images of the same vehicle from different cameras were manually matched as the ground-truth for testing. Due to the difference of surveillance areas and missing detection, some vehicles only appear in one camera or two, especially at the beginning or the end of videos. Finally, 883 vehicles from camera 2 and 899 vehicles from camera 3 were found in camera 1.

The results of the field validation are summarized in Table 4. All the testing accuracies reach over 92.5% among the four testing groups that validates the effectiveness of the proposed vehicle identification method. In addition, the re-identification accuracy of test 2 is better than test 1 by 5.4% and the test 4 is better than test 3 by 5.2%. It is due to the fact that camera 1 and camera 3 have close direction for the visual field, which is opposite to that of camera 2. Moreover, seen from the comparison between test 1 and test 3, or test 2 and test 4, the utilization of re-ranking slightly improved the accuracy.

### 6.3. Recognition Results Based on Different Datasets

In order to investigate the vehicle re-identification performance of the proposed method, the public dataset, including the VeRi-776 and the VeRi-Wild, were utilized for testing. The former contains 51,038 images of 776 vehicles, among which 37,781 images were used for training and 13,257 images for testing. The VeRi-Wild contains 416,314 images of 40,671 vehicles, and 277,797 images were used for training and 138,517 images were for testing.

Table 5 shows their performances in field validation. Though VeRi-Wild has the largest amount of vehicle images, the model trained on it is not as good as the model trained on VeRi-776. Additionally, the models trained by the open-sourced datasets did not reach the same level as the model trained by the established dataset. The reasons responsible for their low accuracies are discussed in the next section.

## 7. Discussions of Incorrect Recognition

The discussion of errors helps open an insight into the issues for vehicle tracking along the whole bridge. In this section, the error discussion contains three parts, the vehicle detection errors in the visual field of the same camera, the vehicle re-identification errors among multiple cameras, and the reason for the different performances based on different datasets.

### 7.1. Recognition Results Based on Different Datasets

Figure 16 shows the main reasons that cause misdetection. The numbers in Figure 16 are the serial number of the detected vehicle and the bounding boxes were assigned with different colors to distinguish from each other. The blocked cars, shown in Figure 16a, should be responsible for false identification and this is the inherent defect of the vision-based method. The second one is the instability of detection, shown in Figure 16b. Despite the fact that the Kalman Filter was used to eliminate interference, the significant change in the bounding box can lead to mistakes. The last reason is the multiple boxes of the same object, as shown in Figure 16c. This usually does not last long and will be eliminated by our strategies, but when it happens, it can seriously affect vehicle tracking.

### 7.2. Discussions of Vehicle Re-Identification Errors

In the re-identification module, the input is the output of the detection module, so a query vehicle may not have its corresponding picture in the basic camera but will be forced to choose one, which means that misdetection will inevitably result in false matches. Another reason is the similar appearance of vehicles and samples are shown in Figure 17a. The low definition of surveillance cameras worsens this situation. Furthermore, the vehicles that are in close proximity with the query one will be captured and will interfere with feature extraction, as shown in Figure 17b.

The results between camera 1 and camera 2 are not so desirable. The public dataset usually provides multi-views of vehicle images to restore the vehicle’s appearance and applying multiple cameras that cover various sights as basic cameras may alleviate the problem. In view of the high accuracy between camera 1 and camera 3, a tentative means was applied to simulate multiple basic cameras in Table 6. In test 2, the pictures in camera 3 were relabeled as the label of their corresponding ones in camera 1 and mixed with that, while in test 3, they were input with no label and only worked in the reranking step.

Though camera 3 does not share a similar sight with query pictures, it seems to benefit re-identification a lot if pictures are labeled. However, it is impractical to note pictures manually for industrial application. Moreover, accuracy has improved even if the additional pictures are unlabeled, which shows the effectiveness of reranking.

### 7.3. Discussions of the Difference of Performance among Multiple Datasets

The VeRi-Wild and VeRi-776 contain many more images than the established dataset but does not work as expected. The main reason is that their images have various definitions, as shown in Figure 18, which describes three typical images at the same dots per inch (DPI). It is obvious that VeRi-Wild has the highest definition and allows neural networks to recognize tiny parts, which does not work on our field validation due to the limitation of low definition.

Figure 19 shows the heat maps of the same image obtained from diverse models and reveals another reason that the difference in distribution between training and testing data leads to a low accuracy. In the heat maps, the colors stand for the contribution of the corresponding areas to the identification of the vehicles. Darker color means larger contribution. The barriers and cables of the bridges which hardly appear in the VeRi-776 or VeRi-Wild seriously affect the recognition. When the models focus on the wrong object, wrong predictions are inevitable.

Through the above analysis, the significance of selecting an appropriate dataset similar to the object of study is clearly proved, and the distribution, along with the scale of the dataset, should be considered for better performance.

## 8. Conclusions

In this paper, a YOLOv4 and OSNet-based method for identification of temporal–spatial distribution of vehicle loads on bridges by means of multiple cameras was proposed, and a dataset containing 25,080 images of 1727 vehicles was established. Several re-identification models were adopted to conduct a comparison study with the OSNet, and a re-ranking method was applied to improve performances. Field validation on Jiubao bridge was conducted to verify accuracy of the proposed method. According to the study, some conclusions can be drawn as follows:(1)The combination of the YOLOv4 and a modified IoU-based tracking method realizes the detection and tracking of vehicles on bridges in a single camera, and has an accuracy of 97.7%. In addition, the proposed adaptive lane recognition algorithm improves the location of vehicles precisely without extra considerable computation.(2)In terms of the mAP and Rank-1 indices, the OSNet outperforms the other re-identification models and was chosen to verify our method. The accuracies of the OSNet-based re-identification method in field validation reached over 92.5% and 97.9% for camera 2 and camera 3, respectively, which indicates that vehicles can be precisely re-identified through multiple cameras without overlapped visual fields.(3)With the introduction of the re-ranking method, the improvement in accuracy is 0.5% and 0.3% in camera 2 and camera 3, respectively. Though it only benefits the result slightly, further investigation shows that the effect can be enhanced by inputting more images even if they are unlabeled. The re-ranking method can reduce mistakes in vehicle re-identification, especially between two cameras with different sights.(4)The realization of the proposed method can contribute to the acquisition of the temporal–spatial distribution of vehicles on the whole bridge for precise estimation of vehicle loads.

## Figures and Tables

**Figure 1 sensors-23-05510-f001:**
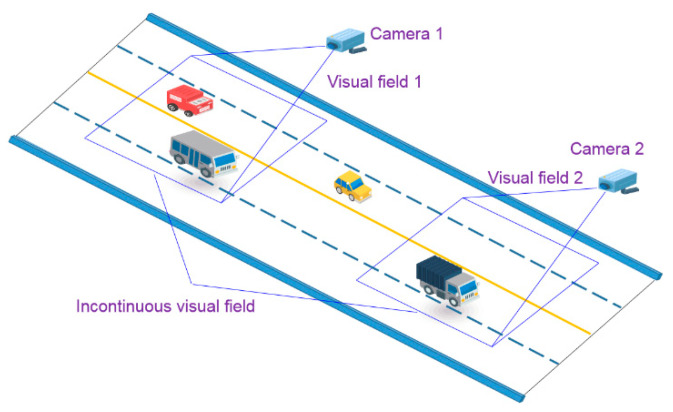
Challenges of computer vision-based vehicle detection.

**Figure 2 sensors-23-05510-f002:**
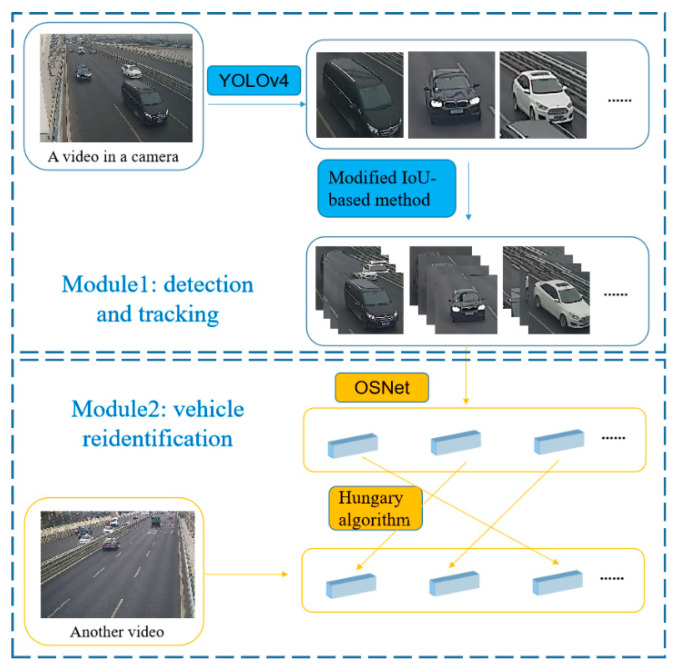
Framework of the proposed method.

**Figure 3 sensors-23-05510-f003:**
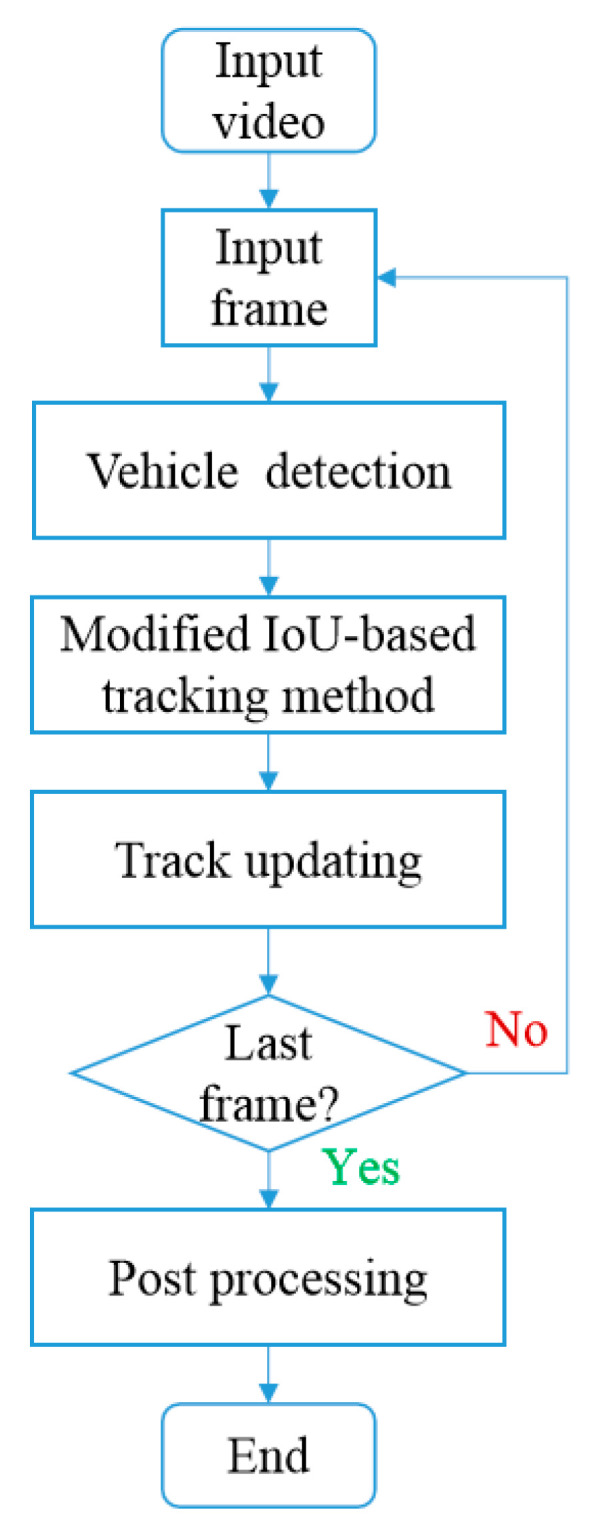
Flowchart of vehicle detection and tracking.

**Figure 4 sensors-23-05510-f004:**
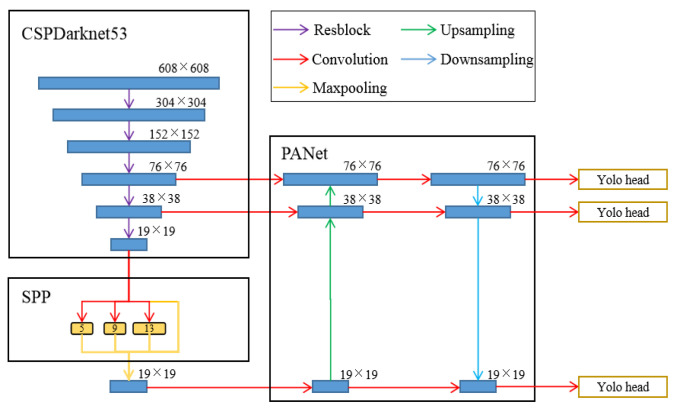
The architecture of the YOLOv4.

**Figure 5 sensors-23-05510-f005:**
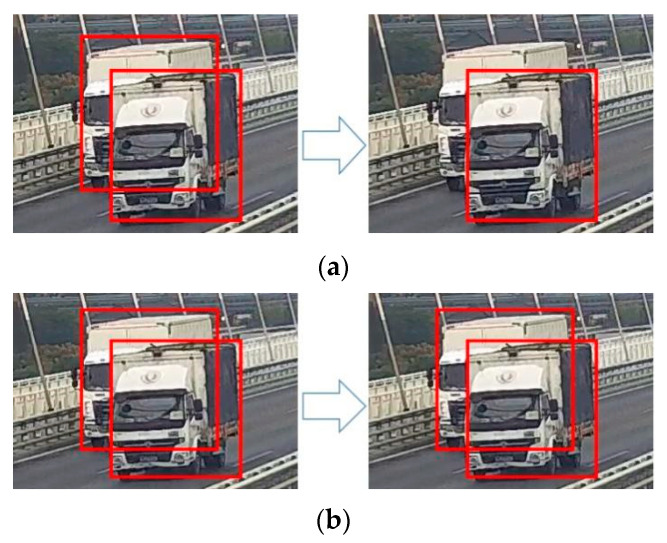
Difference between IoU-NMS and DIoU-NMS algorithm: (**a**) IoU-NMS algorithm; (**b**) DIoU-NMS algorithm.

**Figure 6 sensors-23-05510-f006:**
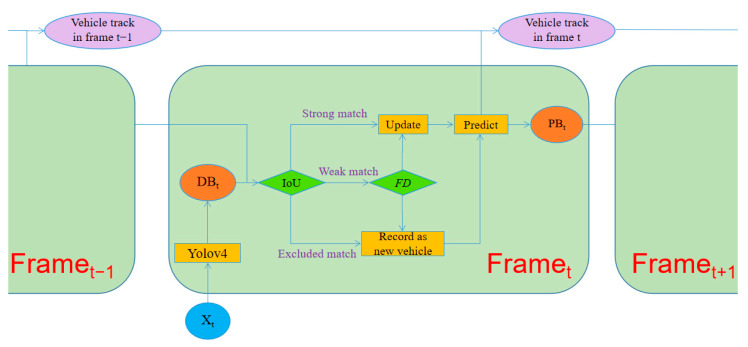
Modified IoU-based tracking method.

**Figure 7 sensors-23-05510-f007:**
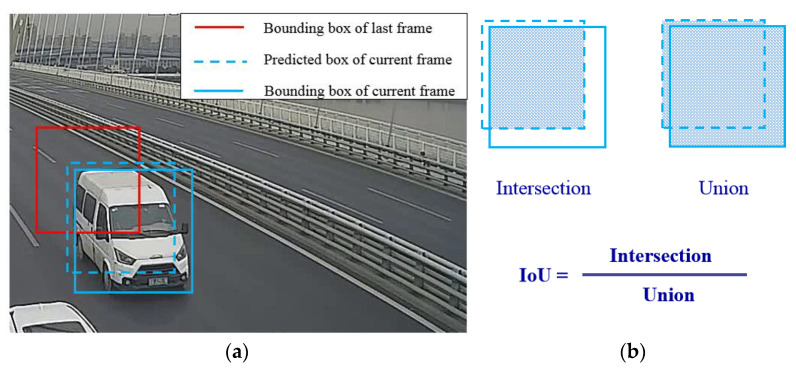
Vehicle detection with the modified IoU-based tracking method: (**a**) the abridged general view of bounding boxes; (**b**) the formula for calculating the IoU.

**Figure 8 sensors-23-05510-f008:**
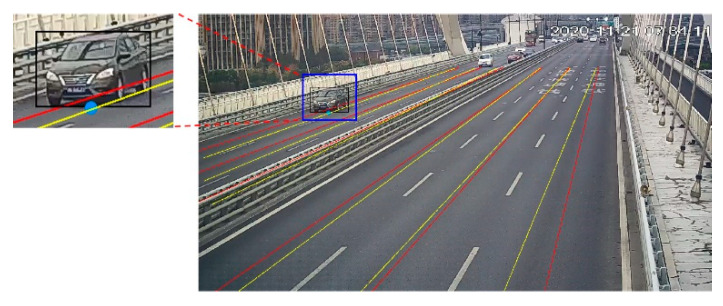
Mismatch between the bounding box and lane.

**Figure 9 sensors-23-05510-f009:**
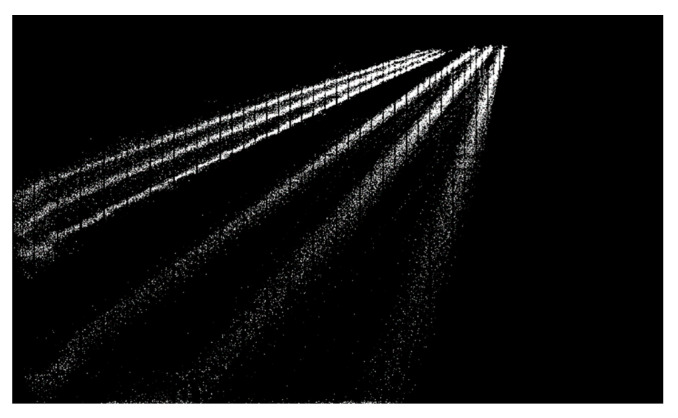
The tracks of the detected vehicles.

**Figure 10 sensors-23-05510-f010:**
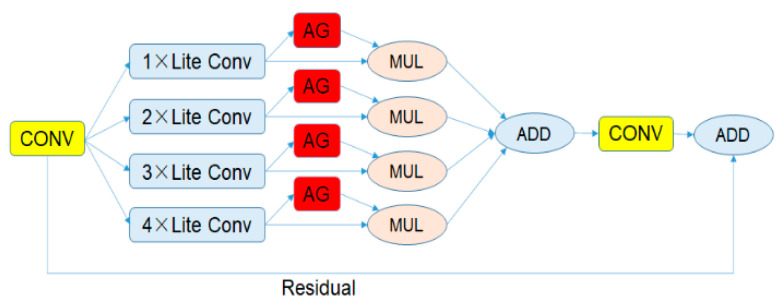
The bottleneck of the OSNet.

**Figure 11 sensors-23-05510-f011:**
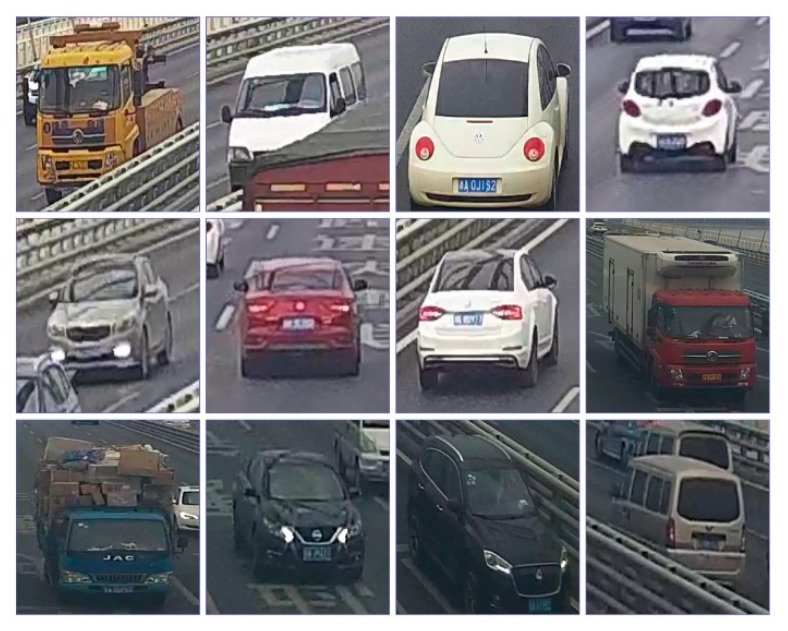
Samples of the established vehicle dataset.

**Figure 12 sensors-23-05510-f012:**
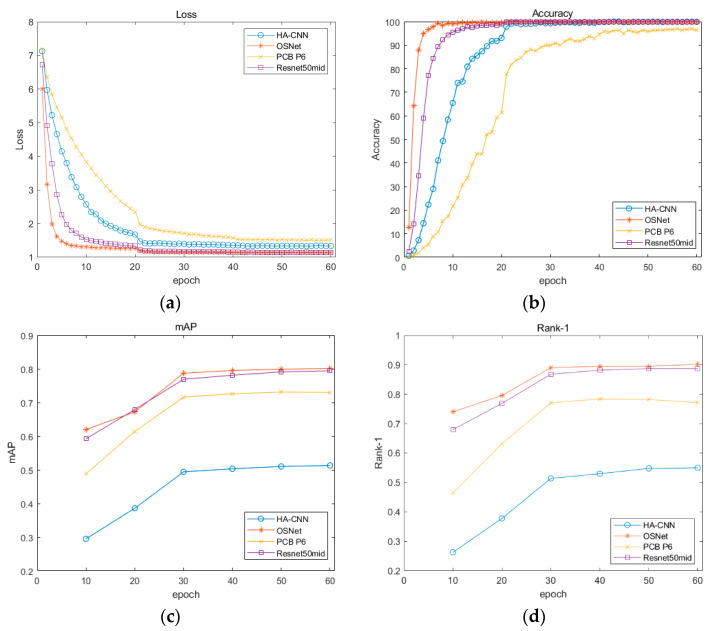
Performance evaluation of the re-identification models: (**a**) loss in training step; (**b**) accuracy in training step; (**c**) mAP in testing step; (**d**) Rank-1 in testing step.

**Figure 13 sensors-23-05510-f013:**
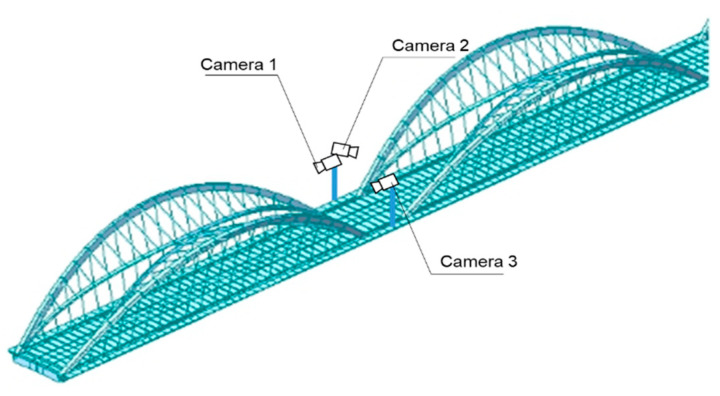
Layout of the cameras on the bridge.

**Figure 14 sensors-23-05510-f014:**
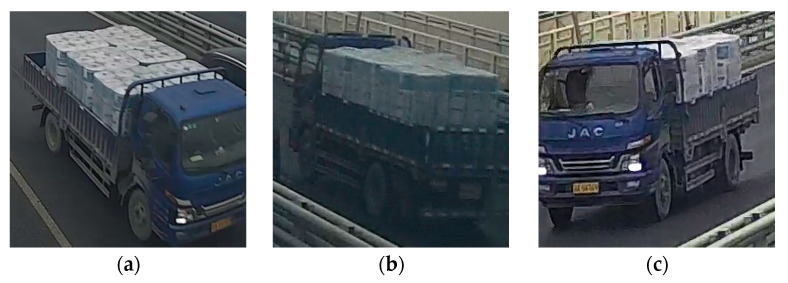
Images of the same truck captured by three cameras: (**a**) camera 1; (**b**) camera 2; (**c**) camera 3.

**Figure 15 sensors-23-05510-f015:**
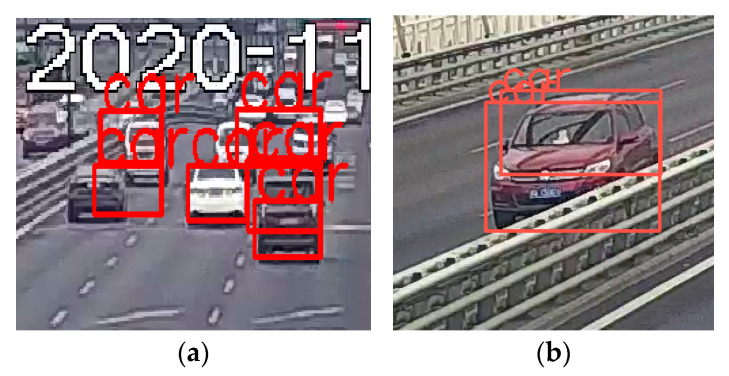
Factors affecting recognition: (**a**) inexact positioning; (**b**) incorrect detection.

**Figure 16 sensors-23-05510-f016:**
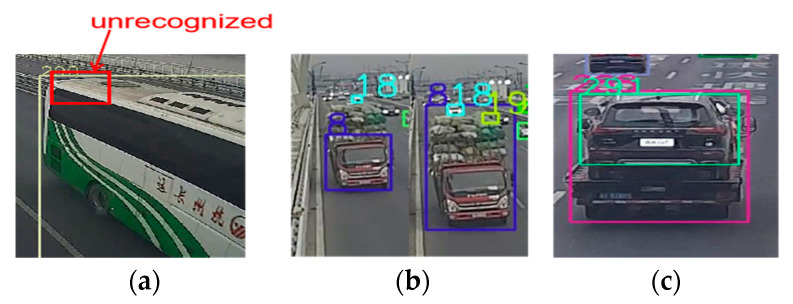
Reasons for misdetection: (**a**) blocked vehicles; (**b**) instability of detection; (**c**) multiple boxes of an object.

**Figure 17 sensors-23-05510-f017:**
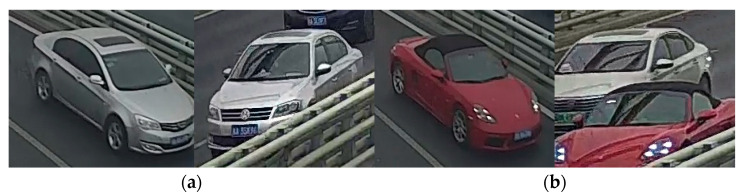
Samples of misidentification: (**a**) the similar appearance of different vehicles; (**b**) a picture interfered by another vehicle.

**Figure 18 sensors-23-05510-f018:**
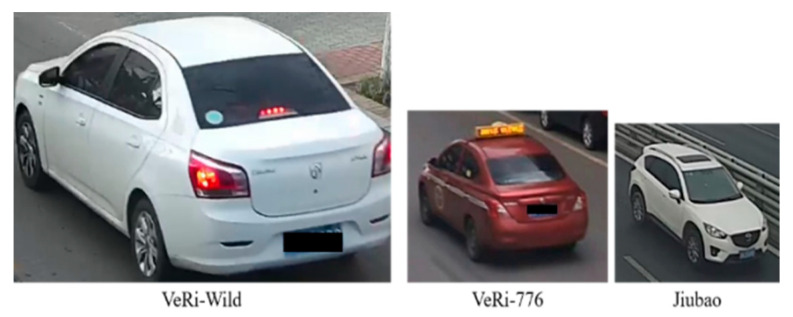
Images in three datasets shown at the same DPI.

**Figure 19 sensors-23-05510-f019:**
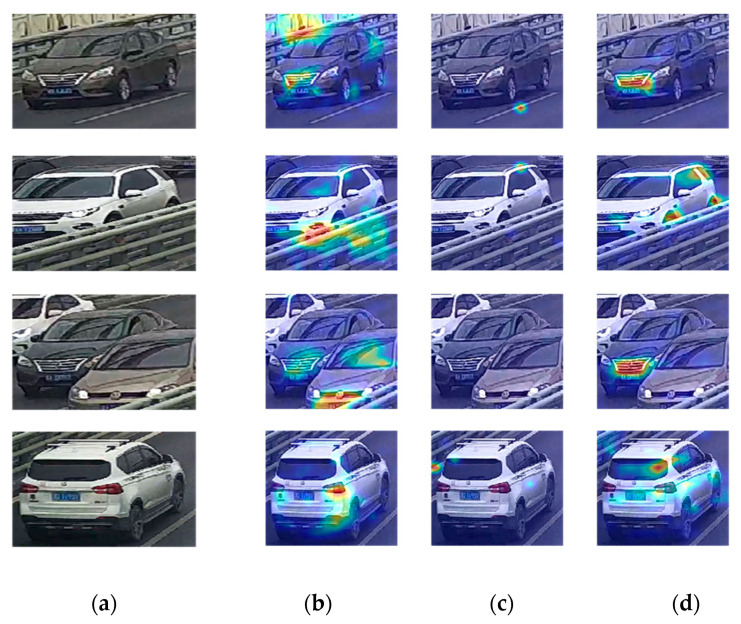
Heatmaps by models on various datasets: (**a**) original images; (**b**) VeRi-776; (**c**) VeRi-Wild; and (**d**) VeRi-Jiubao.

**Table 1 sensors-23-05510-t001:** Hyperparameter for training models.

Size of Input	Max Epoch	Batch Size	Optimizer	Initial Learning Rate	Loss Function
256 × 256	60	32	Adam	0.0003	Triplet Loss + Cross Entropy Loss

**Table 2 sensors-23-05510-t002:** Hardware and software for training.

Item	Version
**Hardware**	CPU: 2 × Intel(R) Xeon(R) Silver4215R CPU @ 3.20 GHz
	GPU: NVIDIA RTX 3090/GDDR5X 24 GB
	RAM: 64 GB
**Software**	Windows 10 Version 1909
	Pytorch 1.7.1 + cu110
	Python 3.8.5
	Opencv 4.4.0
	Torchreid 1.3.3

**Table 3 sensors-23-05510-t003:** Vehicle detection accuracy in video frames from three cameras.

Camera	Correct Detection	Mis-Detection	False Detection	Ground Truth	Accuracy
camera_1	909	14	2	923	98.3%
camera_2	905	23	1	928	97.4%
camera_3	909	7	17	916	97.4%
sum	2723	44	20	2767	97.7%

**Table 4 sensors-23-05510-t004:** Comparison among different cameras.

Test Number	Target Camera	Reranking or Not	Number of Vehicles	Correct Matches	False Matches	Accuracy
1	camera_2	N	883	817	66	92.5%
2	camera_3	N	899	881	18	97.9%
3	camera_2	Y	883	821	62	93.0%
4	camera_3	Y	899	883	16	98.2%

**Table 5 sensors-23-05510-t005:** Comparison among different datasets.

Test Number	Target Camera	Dataset	Number of Vehicles	True Matches	False Matches	Accuracy
1	camera_2	VeRi-776	883	526	357	59.6%
2	camera_3	VeRi-776	899	768	131	85.4%
3	camera_2	VeRi-Wild	883	297	586	33.6%
4	camera_3	VeRi-Wild	899	569	330	63.3%
5	camera_2	Jiubao	883	821	62	93.0%
6	camera_3	Jiubao	899	883	16	98.2%

**Table 6 sensors-23-05510-t006:** Comparison based on different basic cameras.

Test Number	Target Camera	Basic Camera	Re-Relabeling	Number of Vehicles	True Matches	False Matches	Accuracy
1	camera_2	1	/	883	821	62	93.0%
2	camera_2	1+3	Yes	883	852	31	96.5%
3	camera_2	1+3	No	883	843	34	95.5%

## Data Availability

The data that support the findings of this study are available from the corresponding author upon reasonable request.

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
