# Peer review of "Identification and Tracking of Vehicles between Multiple Cameras on Bridges Using a YOLOv4 and OSNet-Based Method"

_sensors, 2023, doi:10.3390/s23125510_

Round 1

Reviewer 1 Report

Upon reading the manuscript entitled "YOLOv4 and OSNet-based Method for Identification of Temporal Spatial-Distribution of Vehicle Loads on Bridges", a thorough review was conducted.

In the paper, a novel method for estimating vehicle loads on bridges through a combination of YOLOv4 and OSNet-based deep learning techniques was proposed, which is an innovative and timely approach. It was noticed that the authors have made considerable efforts to address the issue of tracking vehicles from multiple cameras without overlapping visual fields, which is a challenging problem in the field.

The paper was well-structured, and the literature review section provided a solid background for the proposed method. The research methodology was well-developed, detailed, and sound. A considerable amount of data, which includes 25080 images of 1727 vehicles, was collected and used effectively to train and evaluate the models, indicating a robust methodology.

The paper's experimental design, particularly the field validation experiments, was commendable. It was found that the method's accuracy in single-camera visual fields and multiple-camera fields was impressive, which adds considerable value to the paper. The use of multiple performance metrics, including mAP and Rank-1 indices, further strengthens the study's reliability.

The conclusions drawn in the paper were clear, concise, and well supported by the data. It was particularly noted that the authors made an important contribution by developing an adaptive lane recognition algorithm, which significantly improves the location accuracy of the vehicles.

Nevertheless, some minor points could be improved to enhance the manuscript further. These comments, however, do not detract from the overall quality of the work presented in this paper.

Overall, this study is an important contribution to the field of structural health monitoring using deep learning techniques. Therefore, it is recommended that this paper be accepted for publication.

Reviewer 2 Report

The authors proposed a method based on YOLOv4 and OSNet to identify the temporal-spatial distribution of vehicle loads on bridges. The manuscript is well prepared. The following issues should be strengthened to improve the manuscript.

1. YOLO has several versions, and YOLOv4 is not the latest version. Please explain the reason why the authors selected YOLOv4 rather than other versions.

2. The title is “identification of temporal-spatial distribution of vehicle loads on bridges using a YOLOv4 and OSNet-based method”, however, the authors only detect the passing vehicles using the YOLOv4 and OSNet based method. The temporal-spatial information of each identified vehicle was not extracted. Hence, please re-consider the title carefully.

3. In this study, it seems that the vehicles do not change lane. How about the performance of the method when the vehicle changes the lane?

The language is generally fine.

Reviewer 3 Report

The paper presents the experimental results obtained with the implemented methodology proposed for identification of temporal-spatial distribution of vehicle loads on bridges. The developed algorithm (its various variants) is based on the YOLO and OSNet methods. Several re-identification models were adopted by the Authors to conduct vehicle tracking. The most effective algorithms used for vehicle recognition were indicated. What is of high importance, the discussion on the causes of improper operation of the proposed approach is conducted by the Authors. In the reviewer’s opinion, the paper is worth to be published however the discussion on the presented below comments should by added to the manuscript.

The comment is required regarding comparison of the performance of the two approaches: 1) following the cited publication “Chen et al. [20] established a real-time vehicle detection and counting method based on single shot detection (SSD) and reached an accuracy of 99.3%.”  VS.  2) the Authors’ statement “Experimental results show that the proposed method gets an accuracy of 97.7% in vehicle tracking in the visual field of a single camera” (single shot vs. detection based on a single camera).

Is it feasible an on-line (in real time) application of the proposed approach in case of highly crowded areas? What is the calculation time required to extract the features for all the tracked vehicles?

What is the minimum image resolution that allows for vehicle recognition and tracking?

Are there any limitations regarding the number of vehicles being tracked? What is the minimum distance (or relative distance referenced to the overall dimension) between the vehicles that still allows for their correct detection and tracking?

Minor comments, flaws, stylistic errors:

Abstract: “The estimation of vehicle loads estimation”->”The estimation of vehicle loads”

Abstract: “Nevertheless, how to keep track of vehicles ...” -> ”Nevertheless, keeping track of vehicles ...”

Row 44: “[11]developed” -> “[11] developed” (space required)

Row 80: “distances between the adjacent vehicles are close” -> “distances between the adjacent vehicles are small”   the distance cannot be close, it can be small OR the vehicles can be close one to another

In the whole paper: I would recommend that you should substitute the word “trick/tricks” as it is mostly used in common, spoken language rather than in a technical/official document. Some examples to substitute: “approaches”, “methods”.

Row 223: “Red lines are”->” Red lines shown in Fig. 8 are”

Row 278: “Every vehicle picture has a couple of pictures” -> “Every registered vehicle has a couple of pictures”

Row 279: the expression “and Precision_i is the precision up to when the i_th one of them appears.” should be paraphrased as in its present form is not clear.

Row 330: “balance” -> “balances”

Row 348: the expression “every element minus the minimum value for its column” should paraphrased
